# A Novel High *Q* Lamé-Mode Bulk Resonator with Low Bias Voltage

**DOI:** 10.3390/mi11080737

**Published:** 2020-07-29

**Authors:** Tianyun Wang, Zeji Chen, Qianqian Jia, Quan Yuan, Jinling Yang, Fuhua Yang

**Affiliations:** 1Institute of Semiconductors, Chinese Academy of Sciences, Beijing 100083, China; tywang@semi.ac.cn (T.W.); chenzeji@semi.ac.cn (Z.C.); jiaqianqian18@mails.ucas.ac.cn (Q.J.); fhyang@semi.ac.cn (F.Y.); 2Center of Materials Science and Optoelectronics Engineering, University of Chinese Academy of Sciences, Beijing 100049, China; 3State Key Laboratory of Transducer Technology, Shanghai 200050, China

**Keywords:** Lamé-mode, resonators, MEMS, quality factor

## Abstract

This work reports a novel silicon on insulator (SOI)-based high quality factor (*Q* factor) Lamé-mode bulk resonator which can be driven into vibration by a bias voltage as low as 3 V. A SOI-based fabrication process was developed to produce the resonators with 70 nm air gaps, which have a high resonance frequency of 51.3 MHz and high *Q* factors over 8000 in air and over 30,000 in vacuum. The high *Q* values, nano-scale air gaps, and large electrode area greatly improve the capacitive transduction efficiency, which decreases the bias voltage for the high-stiffness bulk mode resonators with high *Q*. The resonator showed the nonlinear behavior. The proposed resonator can be applied to construct a wireless communication system with low power consumption and integrated circuit (IC) integration.

## 1. Introduction

Nowadays, wireless communication systems are developing towards higher frequency, narrow channel, multiband and multimode [1,2], which requires high performance, high integration, and low power consumption resonators as time reference devices [3,4]. So far, the quartz crystals are widely used in wireless communications. However, the quartz crystals have difficulties with miniaturization, on-chip integration, and impact resistance, which limit their application [5]. Silicon-based micro-electro-mechanical system (MEMS) resonators have attracted great attention for their advantages in high performance, small size, low cost, good IC compatibility, and low power consumption [6,7,8]. Enormous efforts have been made in recent years to demonstrate the high-quality factors (*Q*) of the bulk acoustic wave (BAW) mode resonator in comparison to flexural beam resonators [9]. However, the electrostatic actuation/detection of such stiff mechanical modes requires considerably high bias voltages [10,11,12,13,14,15,16,17]. The high voltage limits the practical applications of BAW resonators [18]. A typical Lamé-mode resonator with an extremely high *Q* factor of 7.5 × 10^5^ at a resonance frequency of 12.9 MHz requires a driving voltage of 100 V [10]. Increasing the capacitive area, shrinking the air gap between the resonator and the electrodes or utilizing different detection methods were effective in reducing the bias voltage, but these routines were often restricted by the fabrication technology [19,20]. A Lamé-mode resonator with thin air gap of 50 nm was excited with a low voltage of 2.5 V into vibration at 17.6 MHz, but its *Q* factor was only 8000 in vacuum [19]. Additionally, a capacitively actuated and piezoresistively detected Lamé-mode resonator can vibrate at 2.2 MHz with low bias voltage of 3 V, yet its *Q* factor is 6771 at atmosphere [20]. The comparison between state-of-the-art works and this work is presented in Table 1. Reducing the bias voltage while maintaining high *Q* factor is essential for achieving an IC-integrable high-performance MEMS resonator.

In this work, a novel silicon on insulator (SOI)-based Lamé-mode bulk acoustic resonator vibrating at a low bias voltage is presented. A reliable SOI-based fabrication process was developed to produce the bulk mode resonator with 70 nm air gap between the resonators and the electrodes. The nano-scale air gaps, high *Q* factor of the resonator, and large electrode area were favorable for achieving efficient capacitive transduction with a low direct current (DC) bias voltage. The transmission performance of the resonator operating in air and in vacuum indicates that the *Q* value is mainly determined by air damping when the resonator is operating in air. The nonlinearities of the device were also experimentally observed.

## 2. Design and Fabrication

The resonator is designed to vibrate in Lamé-mode. The simulated mode shape with COMSOL is shown in Figure 1. Lamé-mode is a bulk acoustic wave (BAW) mode which has low air damping and low thermal elastic damping losses, and therefore is expected to achieve high *Q* factor. In this mode, the edges of the square plate deform in the antiphase, while the volume of the plate is preserved. The length of the square is designed to be 75 µm. Capacitive transduction, which can offer better frequency-*Q* products [21,22,23], is used for exciting and sensing the mechanical resonance signal of the resonator. Four electrodes are placed parallel to the four sidewalls of the square plate for a large transduction area, and four nano-scale air gaps are designed to enhance the mechanical-electrical transduction efficiency. The anchor beams are located at the mode nodal points near each corner of the plate for minimum energy losses through the anchors.

The resonant frequency *f*_0_ is determined by the effective spring constant *k*_eff_ and the effective mass *m*_eff_, which can be expressed using the following equation [24]:(1)$$f_{0}=\frac{1}{2\pi }\sqrt{\frac{k_{\mathrm{eff}}}{m_{eff}}}$$

The effective spring constant and the effective mass of Lamé-mode can be approximated as [25]:(2)$$k_{\mathrm{eff}}=\pi^{2}Gh$$
(3)$$m_{\mathrm{eff}}=\frac{1}{2}\rho hL^{2}$$
where *G* represents the shear modulus, *h* is the thickness of the resonator, *ρ* is the density of silicon, and *L* is the length of the square. Equations (1)–(3) can be combined into the equation below [26]:(4)$$f_{0}=\frac{1}{\sqrt{2}L}\sqrt{\frac{G}{\rho }}$$
and the shear modulus can be expressed as:(5)$$G=\frac{E}{2(1+\nu )}$$
for the single crystal silicon, where *E* = 180 GPa, *ρ* = 2330 kg/m^3^, and *ν* = 0.29; according to Equation (4), the calculated resonance frequency of the square resonator with *L* = 75 µm is around 51.6 MHz, which is verified by COMSOL simulation.

The *Q* factor of a resonator is a dimensionless parameter and can be defined as the ratio between the total stored energy and the average energy loss per cycle [16]. Several dissipation mechanisms contribute to the total value of *Q* [27]:(6)$$\frac{1}{Q_{\mathrm{total}}}=\frac{1}{Q_{\mathrm{anchor}}}+\frac{1}{Q_{\mathrm{TED}}}+\frac{1}{Q_{\mathrm{surface}}}+\frac{1}{Q_{\mathrm{air}}}+\frac{1}{Q_{\mathrm{Akhiezer}}}+\frac{1}{Q_{\mathrm{other}}}$$

For the Lamé-mode resonator, the most important dissipation mechanisms include air damping, surface loss, thermo-elastic dissipation (TED), and anchor losses [28]. For resonators operating in atmosphere, air damping is the dominant dissipation. When operated in vacuum, the losses due to air damping can be significantly reduced. TED of the Lamé-mode resonator in very high frequency (VHF) and ultra high frequency (UHF) range is also low, since the volume of the structure is conserved during vibration [25]. Anchor loss is commonly understood to be determined by energy dissipation through the anchor point structure to the substrate when the resonator vibrates [29,30]. Optimizing the structure and size of the anchor beam can reduce the energy dissipation. Furthermore, the anchor loss can be simulated with the perfectly matched layer (PML) method [31] by building an efficient model of energy losses through the substrate in COMSOL, as shown in Figure 2. The anchor dimensions can be optimized to reduce the anchor loss, and the quality factor is extracted as 7.06 × 10^5^.

The small-signal equivalent electrical model of Lamé-mode resonator can be expressed using the Butterworth van Dyke (BVD) circuit model, as shown in Figure 3.

The motional resistance *R*_m_, motional inductance *L*_m_, motional capacitance *C*_m_, and electromechanical coupling coefficient *η* for the Lamé-mode resonator can be expressed using the following equations [32]:(7)$$R_{m}=\frac{\sqrt{k_{\mathrm{eff}}m_{\mathrm{eff}}}}{Q\eta^{2}},L_{m}=\frac{m_{\mathrm{eff}}}{\eta^{2}},C_{m}=\frac{\eta^{2}}{k_{\mathrm{eff}}}$$
(8)$$\eta =\frac{2\varepsilon_{0}Lh}{g^{2}}V_{\mathrm{DC}}$$
where *ε*_0_ is the permittivity, *g* is the spacing gap, and *V*_DC_ is the bias voltage. Combining Equation (7) and Equation (8), the equivalent output electrical resistance of the resonator can be expressed by [33]:(9)$$R_{m}=\frac{\sqrt{k_{\mathrm{eff}}m_{\mathrm{eff}}}g^{4}}{Q\varepsilon_{0}{}^{2}L^{2}h^{2}V_{\mathrm{DC}}{}^{2}}\propto \frac{g^{4}}{Q}$$

It can be seen that ultra-small capacitive gaps and high *Q* are required to reduce the equivalent motional resistance of the MEMS capacitive resonators.

For a parallel plate capacitor, the actuation force can be expressed by [24]:(10)$$F=\frac{1}{2}(\frac{\partial C}{\partial x})V^{2}$$
where *V* is the applied voltage, *x* is the displacement of resonator, *C* = *ε*_0_*A*/*g* is the capacitance of the parallel plate, and *A* is the transduction area, respectively. For a two-port configuration, the input voltage can be written as the sum of a DC bias *V*_DC_ and an alternating current (AC) signal *V*_AC_
*=* |*V*_AC_| cos*(ωt)* at resonant frequency, and the actuator force can be expressed as [20]:(11)$$F=V_{\mathrm{DC}}\left|V_{\mathrm{AC}}\right|\frac{\varepsilon_{0}A}{g^{2}}\cos(\omega_{0}t)=F_{0}\cos(\omega_{0}t)$$

The motional current is given by [19]:(12)$$i_{m}=V_{\mathrm{DC}}(\frac{\partial C}{\partial x})\dot{x}$$
and the maximum displacement at resonance can be expressed as [20]:(13)$$x_{\max}=\frac{F_{0}Q}{m_{\mathrm{eff}}\omega_{0}{}^{2}}$$

Substituting Equation (10)–(12), the motional current can be expressed by [20]:(14)$$i_{m}=\frac{\left|V_{\mathrm{DC}}{}^{2}\right|\left|V_{\mathrm{AC}}\right|Q\varepsilon_{0}{}^{2}A^{2}}{m_{\mathrm{eff}}\omega_{0}g^{4}}$$

It can be seen from Equation (14) that in the capacitive resonator, the sensing of mechanical vibration is limited by the transduction gap, transduction area, and *Q* factor. To reduce the bias voltage while maintaining strong sensing signal, a small gap, high *Q* factor, and large transduction area are needed.

A simple and reliable fabrication process was developed, as illustrated in Figure 4. The silicon on insulator (SOI) wafer with a 2 µm-thick low-resistivity single-crystal-silicon (SCS) device layer, a 1 µm-thick oxide layer, and a 300 µm-thick silicon handling layer was employed to batch fabricate the proposed resonators; an approximately 1.2-µm-thick SiO_2_ layer is grown by plasma enhanced chemical vapor deposition (PECVD) as the dielectric layer and the hard mask for silicon etching. Then, the resonators are patterned by inductively coupled plasma (ICP) dry etch, and a 70 nm gap is defined by sacrificial thermal SiO_2_ layer. The grounding square hole arrays are fabricated by the ICP etching and filled with polysilicon—this can ensure extremely low feedthrough signal in the device. Subsequently, a 2 µm-thick heavily doped low pressure chemical vapor deposition (LPCVD) polysilicon is deposited and patterned to form the electrodes. The Au/Cr electrode pads are produced by e-beam evaporation and the lift-off process. Finally, the devices are released in a 49% concentrated HF solution. Figure 5 demonstrates the SEM images of the fabricated resonator.

The frequency responses of the fabricated resonators are tested by the measurement setup shown in Figure 6. A radio frequency (RF) probe station was employed, a bias voltage *V*_DC_ was directly applied to the resonator using the DC probe, and the substrate wafer was grounded to reduce the parasitic signal. A 0 dBm AC driving signal from the network analyzer was applied to the Cr/Au electrode pads using the AC probe of the RF probe station. A low pressure of 0.08 mbar was provided for measurement in vacuum.

## 3. Results and Discussions

The frequency responses of the fabricated Lamé-mode resonator operating in air and in vacuum are shown in Figure 7. The measured resonant frequency is around 51.3 MHz, which corresponds well with the calculated value from Equation (4). The resonator was driven into vibration at a low bias voltage of 3 V, and a high signal-to-noise ratio over 25 dB was obtained. The Lamé-mode resonator exhibits a *Q* value of 8150 in air and 34,200 in vacuum. The dramatic enhancement of *Q* values in vacuum indicates that the air damping is the dominant energy dissipation. The resonator can effectively suppress feedthrough, which is convenient for the extraction of resonance signals.

To better understand the effect of air damping on the Lamé-mode resonator, a finite element simulation based on squeezed film damping is conducted [34]. A Reynold’s equation is used to model the squeezed film between the resonator and electrodes [35]:(15)$$p_{a}(\frac{\partial^{2}(\delta p)}{\partial y^{2}}+\frac{\partial^{2}(\delta p)}{\partial z^{2}})-\frac{12\eta_{\mathrm{eff}}}{g^{2}}\frac{\partial (\delta p)}{\partial t}=\frac{12\eta_{\mathrm{eff}}p_{a}}{g^{3}}\frac{\partial (u(y,t))}{\partial t}$$
where *p*_a_ is the ambient pressure, *u*(*y*,*t*) represents the deformations of the sidewalls, *δ*_p_ is the pressure changes inside the gap, and *η*_eff_ is the effective viscosity. y and z axes are defined along the length and thickness of the resonator. The boundary condition *δ*_p_ = 0 is applied to the top and bottom surfaces of the gap, since the pressure here is equal to ambient pressure [34]. At the side edges of the gap, the pressure gradient should be zero due to the anchors position:$\;\frac{\delta_{p}}{\partial x}=0$ [34]. Then, *δ*_p_ can be calculated with the boundary conditions described above, thus the energy loss *E*_air_ due to the squeezing film damping can be computed. The maximum stored energy *E*_store_ can be obtained by integrating the elastic potential energy over the resonator volume, the quality factor of the resonator due to the air damping can be estimated as follows:(16)$$Q=2\pi \frac{E_{\mathrm{store}}}{E_{\mathrm{air}}}$$

The calculated *Q* factor due to air damping is 19,997 and 7.69 × 10^7^ at atmosphere and at a low pressure of 0.08 mbar, respectively, indicating that air damping dominates the resonant behavior of the resonator vibrating in air, and a vacuum package is needed for high-end resonators. In addition, the *Q* values measured at atmosphere and in vacuum are smaller than the simulated ones, indicating that there are other sources of energy loss.

The electrical parameters of the resonator can be estimated based on the insertion loss using the following equation [36]:(17)$$R_{m}=50(10^{\frac{IL_{\mathrm{dB}}}{20}}-1),L_{m}=\frac{QR_{m}}{\omega_{0}},C_{m}=\frac{1}{\omega_{0}QR_{m}}$$
where *IL*_dB_ is the insertion loss of the transmission and its unit is in decibels (dB). Table 2 summarizes the calculated and measured electrical parameters for the Lamé-mode resonators. The measured *R*_m_ of the Lamé-mode resonator with an applied bias *V*_DC_ = 3 V in vacuum is 293.8 kΩ, which indicates that both the nano-scale air gap and high *Q* factor contributes to the reduced motional resistance.

For MEMS resonators, a nonlinear effect often occurs in the resonators with small stiffness, such as a beam resonator vibrating in the flexural mode [37]. For the bulk mode resonators with high-stiffness, such as the Lamé-mode resonator, nonlinear vibration seldom takes place. However, when a large driving force is applied, large vibration amplitude will cause frequency hysteresis [38].

For the resonator devices, the nonlinear effect is not desired in many applications. However, for some requirements, such as the MEMS oscillator, of which the power handling capabilities are limited by the small size of the MEMS resonator, it is usually necessary to drive the device into a nonlinear regime to achieve a sufficient signal-to-noise ratio and performance [39]. Therefore, an in-depth study on the nonlinearity of the resonator is important.

The frequency responses of the Lamé-mode resonator driven by different bias voltages are presented in Figure 8. When the driving voltage *V*_DC_ is increased, the resonance frequency decreases due to the frequency tuning effect. The motional resistance and the *Q* factor are slightly improved. However, the effect of nonlinearity is prominent when the bias voltage *V*_DC_ goes beyond 7 V.

In order to study the nonlinear effect, higher-order terms for the stiffness constant are introduced and the dynamic response of nonlinear vibration is [40]:(18)$$m_{\mathrm{eff}}\frac{\partial^{2}x}{\partial t^{2}}+\gamma \frac{\partial x}{\partial t}+k_{1}x+k_{2}x^{2}+k_{3}x^{3}=F_{0}\cos(\omega_{0}t)$$
where γ is the damping coefficient, *k*_1_, *k*_2_, and *k*_3_ are the equivalent linear, quadratic, and cubic spring constants, respectively. In symmetrical structures such as the Lamé-mode resonator presented in this work, *k*_2_ can be ignored [41]. The frequency change Δ*f* due to the nonlinearity of the stiffness constant can be solved from Equation (18), and its approximate solution can be expressed by [42]:(19)$$\Delta f=\kappa x_{\max}^{2}$$
where *x*_max_ is the amplitude of the resonator and *κ* is the coefficient associated with the nonlinear spring constant [42]:(20)$$\kappa =\frac{3k_{3}}{8k_{1}}f_{0}-\frac{5k_{2}^{2}}{12k_{1}^{2}}f_{0}$$

It can be seen from Equation (19) that when the constant *κ* is positive, the nonlinearities cause the resonance peak to bend towards a higher frequency, and when the constant *κ* is negative, the resonance shifts to a lower frequency.

For electrostatic MEMS resonators, the equivalent stiffness is determined by both mechanical stiffness and electrostatic stiffness, so that the stiffness constants can be expressed by: *k*_1_
*= k*_1m_
*+ k*_1e_, *k*_3_
*= k*_3m_
*+ k*_3e_, where *k*_1m_*, k*_1e_ represent the mechanical term and electrostatic term of linear stiffness constant, *k*_3m_, *k*_3e_ represents the corresponding terms of the cubic stiffness constant. Electrostatic nonlinearities are caused by capacitive transduction, and often lead to a spring softening effect [43]. On the other hand, mechanical nonlinearities can be classified as two types: geometrical effects and material effects. For bulk mode resonators, material effects dominate the mechanical nonlinearities [44]. A nonlinear shear modulus is introduced:(21)$$G=G_{0}+G_{1}\gamma +G_{2}\gamma^{2}$$
where *G*_0_, *G*_1_ and *G*_2_ are the linear, first and second order correction terms of shear modulus, respectively. The stiffness constants caused by material effects for the Lamé-mode resonator can be calculated using the following expression [45]:(22)$$k_{1m}=\pi^{2}G_{0}h,k_{2m}=0,k_{3m}=\frac{9\pi^{4}G_{2}h}{4L^{2}}$$

Furthermore, the linear mechanical spring constant *k*_1m_ can be obtained from the experimental data. Ignoring the influence of the higher-order stiffness constant, the resonance frequency *f*_0_ and bias voltage *V*_DC_ has the following relationship due to the electrical softening effect [45]:(23)$$f_{0}=\frac{1}{2\pi }\sqrt{\frac{k_{1m}-k_{1e}}{m_{\mathrm{eff}}}}=\frac{1}{2\pi }\sqrt{\frac{k_{1m}-\frac{V_{\mathrm{DC}}^{2}\varepsilon_{0}A_{\mathrm{total}}}{g^{3}}}{m_{\mathrm{eff}}}}$$
where *A*_total_ is the total area of the resonator electrodes, and the effective mass can be obtained by Equation (3). By fitting the experimental data with Equation (23), the fitted linear mechanical stiffness constant *k*_1m_ is 1.38 ×10^6^ N/m, coinciding with *k*_1m_ calculated by Equation (22). The measured data and fitted curve for the frequency change versus the bias voltage *V*_DC_ are presented in Figure 9.

The cubic spring constant *k*_3_ cannot be experimentally measured, yet it is related to the coefficient *κ* by Equation (20). According to Equation (19), a linear fitting of Δ*f* versus *x*_max_^2^ measured with AC driving voltage range from −3 dBm to 3 dBm was performed with the bias voltage *V*_DC_ = 10 V. The result is presented in Figure 10. The nonlinear parameter *κ* extracted was −1.44 × 10^20^ Hz/m^−2^. Therefore, *k*_3_ can be calculated as −1.03 × 10^19^ N/m^−3^. For resonators with nano-scale gaps, since *k*_3e_ is inversely proportional to g^5^ [43], the electrical nonlinearity should play a more important role than the mechanical nonlinearity and results in the spring soften exhibited by negative *k*_3_.

According to Equation (22), the nonlinear shear modulus *G*_0_ and *G*_2_ values of the Lamé-mode resonator can be extracted. Table 3 presents the comparison between extracted shear modulus and the reported ones.

In the oscillator application of MEMS resonator, nonlinearity will affect its power handling capability. For an ideal oscillator model, Leeson’s equation can be used to characterize the effect of nonlinearity on phase noise [46]:(24)$$L(\Delta f)=10\log(\frac{kTQ}{\pi E_{\mathrm{stored}}f_{0}}+\frac{kTf_{0}}{4\pi E_{\mathrm{stored}}Q\Delta f^{2}})$$
where *L*(Δ*f*) is the phase noise-to-carrier ratio, *k* is Boltzmann’s constant, *T* is the ambient temperature, *E*_stored_ is the energy stored in the resonator. The second term in Equation (24) represents 1/*f*^2^ noise, which can be effectively reduced by increasing *E*_stored_ and *Q*. For the Lamé-mode resonator described in this work, the bulk mode with high stiffness ensures a large stored energy and high *Q* factor.

Therefore, the impact of nonlinearity can be greatly reduced, and the high performance of the oscillator can be achieved.

## 4. Conclusions

A novel Lamé-mode square resonator was developed in this work. The Lamé-mode resonators have high *Q* factors: over 8000 in air and over 30,000 in vacuum. The high *Q* values, nano-scale gaps, and large electrode area greatly improve the capacitive transduction, and make it possible to drive the resonator into vibration at resonance frequency of 51.3 MHz with a voltage as low as 3 V. The nonlinearity of the Lamé-mode square resonator was studied, and the nonlinear parameters were extracted from the measured data to characterize. Such a high-*Q* resonator with low bias voltage has the potential to be utilized in building high-end MEMS oscillators and filters.

## Figures and Tables

**Figure 1 micromachines-11-00737-f001:**
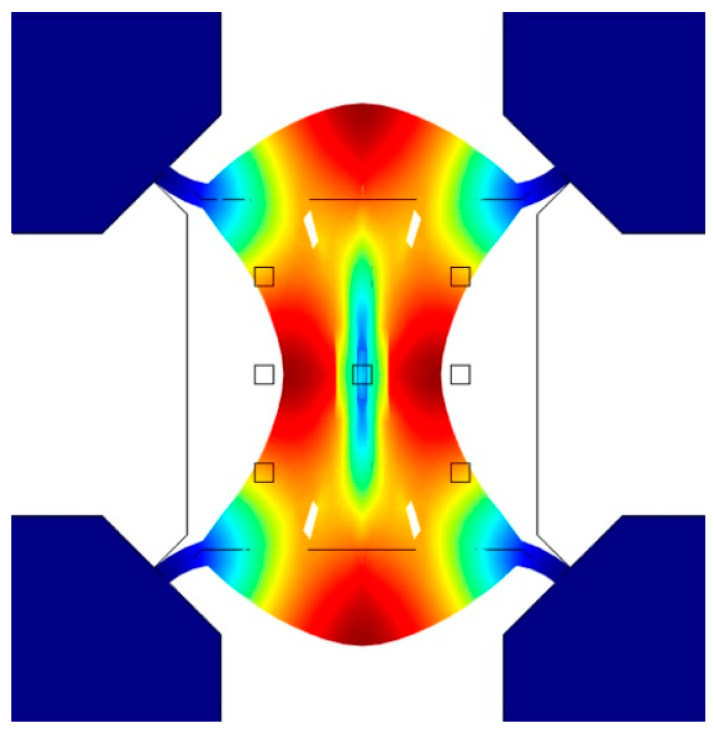
Simulated mode shapes of Lamé-mode resonator.

**Figure 2 micromachines-11-00737-f002:**
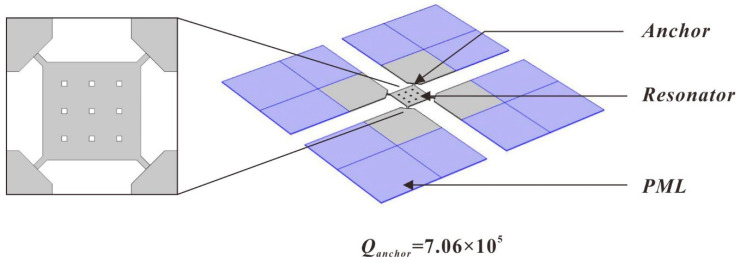
Simulation model of anchor loss using the perfectly matched layer (PML) method in COMSOL.

**Figure 3 micromachines-11-00737-f003:**
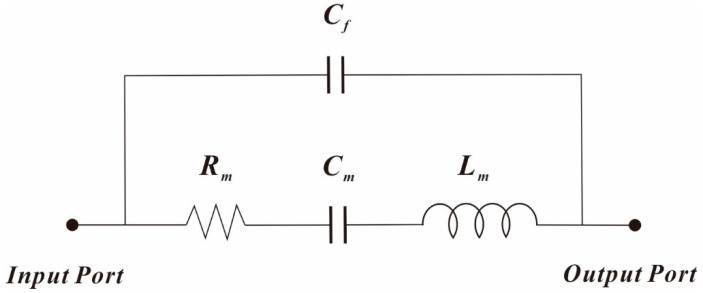
Small-signal equivalent electrical model of the Lamé-mode resonator.

**Figure 4 micromachines-11-00737-f004:**
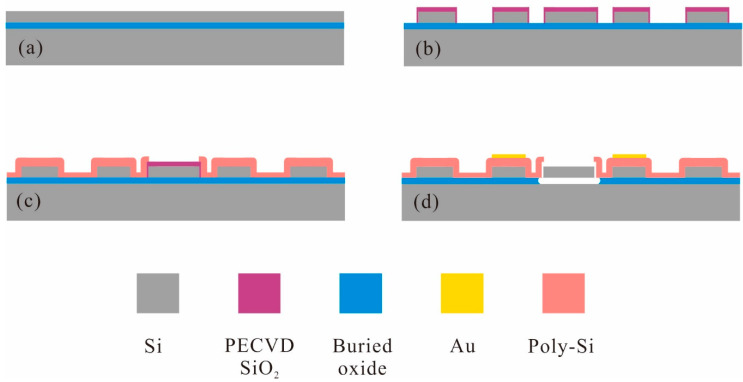
Fabrication process for the Lamé-mode resonator.

**Figure 5 micromachines-11-00737-f005:**
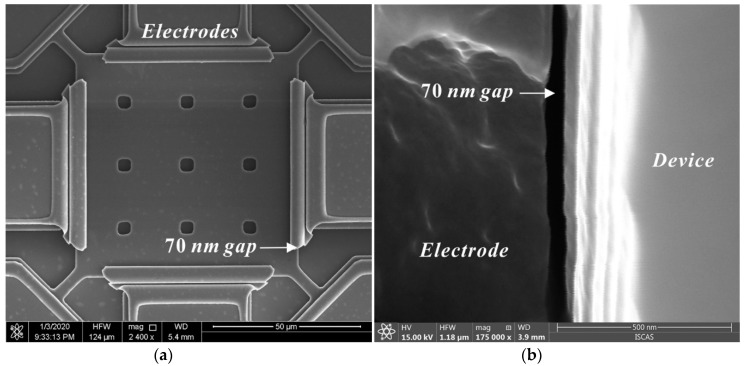
SEM picture of the fabricated Lamé-mode resonator (**a**) and the cross-section of the capacitive 70 nm gaps (**b**).

**Figure 6 micromachines-11-00737-f006:**
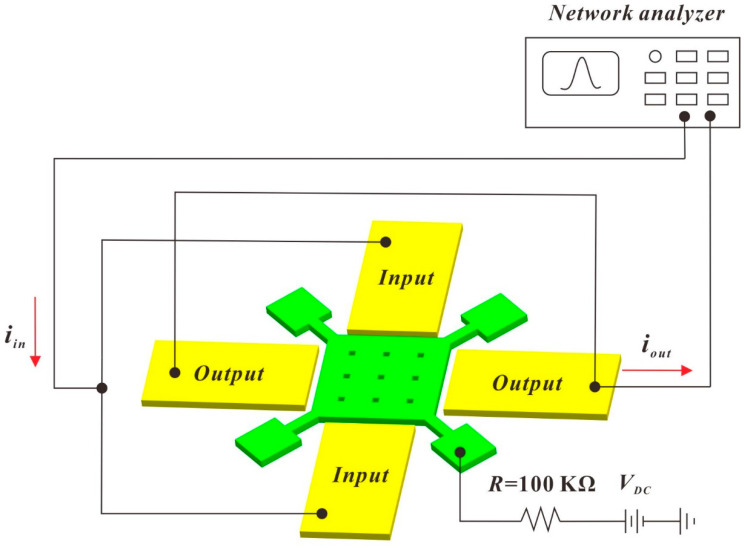
Measurement setup for the Lamé-mode resonator.

**Figure 7 micromachines-11-00737-f007:**
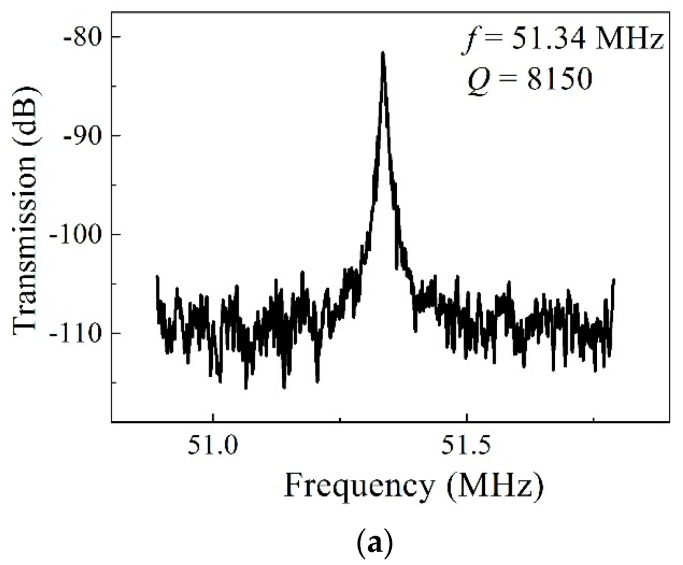
Transmission characteristic curves of the Lamé-mode resonator measured in air (**a**) and in vacuum (0.08 mbar) (**b**).

**Figure 8 micromachines-11-00737-f008:**
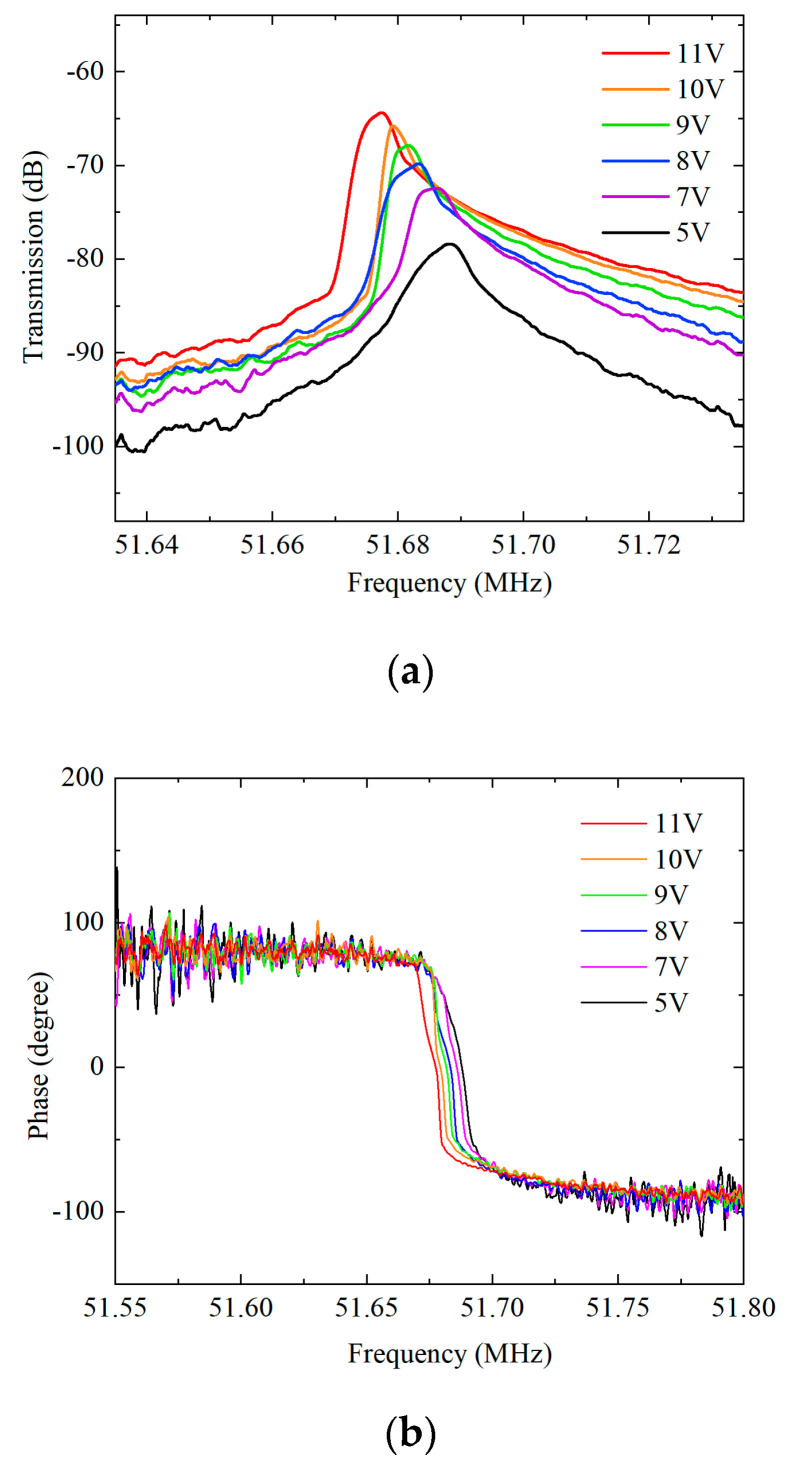
The frequency responses of Lamé-mode resonator operating in atmosphere under different bias voltage in transmission amplitude (**a**) and phase (**b**).

**Figure 9 micromachines-11-00737-f009:**
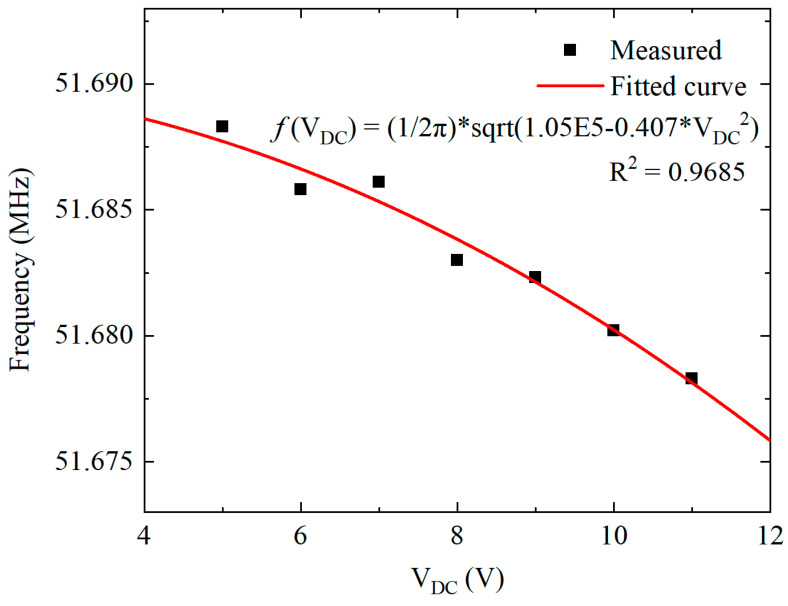
The fitted curve of frequency changes versus the bias voltage of Lamé-mode resonator.

**Figure 10 micromachines-11-00737-f010:**
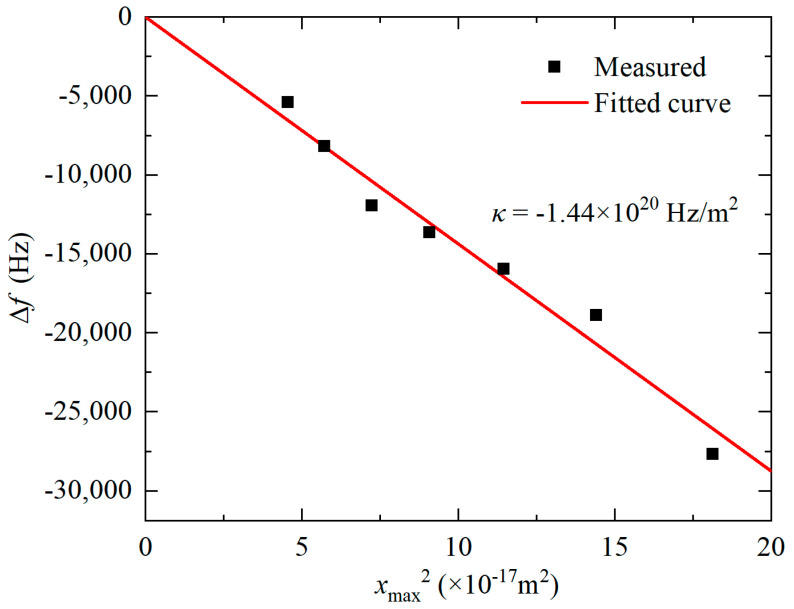
Relationship between ∆*f* and *x*_max_^2^ for the Lame-mode resonator.

**Table 1 micromachines-11-00737-t001:** Comparison between state-of-the-art works and our Lamé mode resonator.

Reference	Resonator	*f* _0_	*Q*	V_DC_	*F × Q*	Gap
Khine, L. [10]	Lamé-mode	12.9 MHz	7.6 × 10^5^	100 V	9.80 × 10^12^	2 μm
Xereas, G. [13]	Lamé-mode	6.89 MHz	3.24 × 10^6^	40 V	2.23 × 10^13^	1.5 μm
Rodriguez, J. [16]	Lamé-mode	10 MHz	2.65 × 10^6^	30 V	2.69 × 10^13^	700 nm
Hamelin, B. [14]	gyroscopic mode SiC disk	5.3 MHz	1.8 × 10^7^	25 V	9 × 10^13^	4.2 μm
Pourkamali, S. [12]	Wine-glass mode disk	149.3 MHz	45.700	17 V	6.82 × 10^12^	100 nm
Yang, J. [15]	SiC Lamé-mode	6.27 MHz	2.0 × 10^7^	15 V	1.25 × 10^14^	5 μm
Daruwalla, A. [17]	Distributed Lamé-mode	50.7 MHz	2.5 × 105	5 V	1.29 × 10^13^	270 nm
Lin, A.T. [20]	Lamé-mode	2.2 MHz	6771	3 V	1.49 × 10^10^	2 μm
Chen, T.T. [19]	Lamé-mode	17.6 MHz	8000	2.5 V	1.41 × 10^11^	50 nm
Our work	Lamé-mode	51.3 MHz	34,200	3 V	1.75 × 10^12^	70 nm

**Table 2 micromachines-11-00737-t002:** Calculated and measured electrical parameters of the Lamé-mode resonator.

Items	*R* _m_	*L* _m_	*C* _m_
Measured	293.8 kΩ	31.2 H	3.09 × 10^−19^ C
Calculated	320.5 kΩ	33.8 H	2.81 × 10^−19^ C

**Table 3 micromachines-11-00737-t003:** Comparison of the extracted nonlinear shear modulus *G*_0_ and *G*_2._

Resonator	*G*_0_ (×10^10^ Pa)	|*G*_2_| (×10^11^ Pa)
Zhu, H. [44]	5.11	2.85
Shao, L. C. [37]	8.73	1.85
Yang, Y. [45]	4.62	8.76
Our work	6.99	4.03

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
