# Peer review of "A Novel High Q Lamé-Mode Bulk Resonator with Low Bias Voltage"

_micromachines, 2020, doi:10.3390/mi11080737_

Round 1

Reviewer 1 Report

1- The title needs to be modified so that the "novelty" of the work is spelled out as opposed to just claiming novelty. In other words in the title, authors should explain what aspect of their work is novel and leave the judgement to the readers.

2- Beyond the title it should also be clear in the abstract what exactly is the novel aspect of the work. Lame-mode bulk silicon resonators have been reported in the literature extensively. Authors have to be specific in how their work is different and back up their claim of novelty with supporting results.

3- An SEM showing the cross-section of the capacitive 70nm gaps is required.

4- A theoretical electrical model for the resonator has to be developed and then compared with the measured results. 

5- The device performance and specifically the motional resistance and the quality factor of the reported resonator appear to be inferior to other similar resonators reported in the past. Authors should explain the possible reasons.

6- Authors claim that the feedthrough signal is extremely small in their resonator. Why is that? How was this achieved? 

7- A picture of the measurement setup is required along with a description of the setup with proper justification.

8- The discussion on nonlinearity is rather incomplete. How does the frequency response change beyond VDC of 11V. Is the bifurcation observed? The nonlinearity observed in the presented frequency responses is not similar to what is shown for other similar work and requires further investigation (the response close to the resonance frequency does not look substantially asymmetric which is the expected hallmark of nonlinearity). 

9- Why is VDC called "driving voltage"? That is usually called the bias voltage in the resonator community.

10- A table comparing the results of this work with similar lame-mode bulk silicon resonators reported in the past has to be added to the paper.

11- The English needs substantial grammatical polishing.

Author Response

Response to the comments of the reviewers

First of all, we are grateful to the referees for their valuable works, which indeed help us improve the quality of our manuscript. We made the further revision to the manuscript according to their comments. All amendments are highlighted as red in the revised manuscript.

Reviewer 1:

Comments and Suggestions for Authors

  1. The title needs to be modified so that the "novelty" of the work is spelled out as opposed to just claiming novelty. In other words in the title, authors should explain what aspect of their work is novel and leave the judgement to the readers.

Our response:

Thank the reviewer for the helpful suggestion.

The novelty of this work is the Lamé-mode bulk resonator which can be excited by a bias voltage as low as 3 V. The title has been changed to "A Novel high Q Lamé-mode Bulk Resonator with Low Bias Voltage" to stress the novelty of the work.

  1. Beyond the title it should also be clear in the abstract what exactly is the novel aspect of the work. Lame-mode bulk silicon resonators have been reported in the literature extensively. Authors have to be specific in how their work is different and back up their claim of novelty with supporting results.

Our response:

Thank the reviewer for the helpful suggestion.

The novelties of this work are the Lame-mode bulk silicon resonator with relatively high Q factor and low bias voltage. The resonator is designed with four anchor beams locating at the nodal points in the plate corner which results in low anchor energy loss and high Q factor. The 70 nm air gap is fabricated to achieve high efficient capacitive transduction and low DC bias voltage of 3V. These merits and novelties of the device are emphasized in the abstract and marked in red.

  1. An SEM showing the cross-section of the capacitive 70nm gaps is required.

Our response:

Thank the reviewer for the helpful suggestion.

The SEM figure of the 70 nm gaps has been added in Figure 5b in the revised manuscript.

  1. A theoretical electrical model for the resonator has to be developed and then compared with the measured results.

Our response:

Thank the reviewer for the helpful suggestion.

The electrical model for the resonator has been provided in Figure 3 and Equation 7, the theoretical and measured value of the key parameters in the model are given and compared in Table 2 in the revised manuscript.

  1. The device performance and specifically the motional resistance and the quality factor of the reported resonator appear to be inferior to other similar resonators reported in the past. Authors should explain the possible reasons.

Our response:

Thank the reviewer for the helpful suggestion.

For the Lamé-mode resonators reported in the past, high bias voltage is required due to the micro-scale transduction gap. Motional resistance is inversely proportional to the square of bias voltage. So the low bias voltage in this work leads to relatively high motional resistance compared with those reported resonators. Nanoscale gap is realized in this work, however, however, the motional resistance of around 300 kΩ is not dramatically decreased. More work will be done to improve the motional resistance by increasing the thickness of the vibrating plate and decreasing the transduction gap.

For the Q factor, the resonator is perforated to facilitate the release processing. The etch holes distort the isovolumetric property of the Lame mode, which results in severe thermal elastic damping, thus leads to Q degradation. Besides, our previous work testified that the slim tether could to reduce anchor loss, however, it is difficult to shrink down the tether proportionally with the device size due to the fabrication process tolerance. Therefore, the anchor loss could be significant. The etch hole distributions as well as the tether design can be further optimized to improve the Q values of the resonator.

  1. Authors claim that the feedthrough signal is extremely small in their resonator. Why is that? How was this achieved?

Our response:

Thank the reviewer for the helpful suggestion.

The feedthrough signal in the resonator was greatly suppressed due to the well-designed grounding structure around the vibrating plate and electrodes. The grounding structure are fabricated with heavily doped polysilicon layer which connected to the underneath grounding substrate through well-distributed vias. This grounding structure ensures extremely low feedthrough signal in the device. The fabrication process of the grounding structure was added in the revised manuscript and marked in red.

  1. A picture of the measurement setup is required along with a description of the setup with proper justification.

Our response:

Thank the reviewer for the helpful suggestion.

The measurement setup has been provided in Figure 6 and the detailed description has been added in the revised manuscript and marked in red.

  1. The discussion on nonlinearity is rather incomplete. How does the frequency response change beyond VDC of 11V. Is the bifurcation observed? The nonlinearity observed in the presented frequency responses is not similar to what is shown for other similar work and requires further investigation (the response close to the resonance frequency does not look substantially asymmetric which is the expected hallmark of nonlinearity).

Our response:

Thank the reviewer for the helpful suggestion.

The preliminary research results on nonlinearity are reported in the manuscript. The square resonators with nanoscale transduction gap were susceptible to breakdown with a bias voltage beyond 11 V. The measured phase spectra have been added, which clearly shows that the device is vibrating in the nonlinear regime. Once the nonlinearity happens, the frequency responses firstly tilt toward lower or higher frequencies. When the excitation force is further increased, the nonlinearity becomes more serious and the transmission signal shows discontinuity due to frequency hysteresis, the bifurcation occurs [i]. Since the resonator in this work was vibrating in the initial phase of nonlinearity, the bifurcation has not been observed. In this work, the mechanism of the nonlinearity has been clarified, the high-order spring constant km3 and nonlinear coefficient κ have been extracted, which are in accordance with the reported work [ii].

  1. Kaajakari, T. Mattila, A. Oja, and H. Seppa, "Nonlinear limits for single-crystal silicon microresonators," J. Microelectromech. Syst., vol. 13, no. 5, pp. 715-724, 2004.https://doi.org/10.1109/jmems.2004.835771
  2. Zhu, C. Tu, and J. E.-Y. Lee, "Material nonlinearity limits on a Lamé-mode single crystal bulk resonator," in 2012 7th IEEE International Conference on Nano/Micro Engineered and Molecular Systems (NEMS), 2012, pp. 457-462: IEEE.

  1. Why is VDC called "driving voltage"? That is usually called the bias voltage in the resonator community.

Our response:

We are sorry for the confusing expression.

We have modified the expression in the revised manuscript and marked in red.

  1. A table comparing the results of this work with similar lame-mode bulk silicon resonators reported in the past has to be added to the paper.

Our response:

Thank the reviewer for the helpful suggestion.

Comparison between state-of-art works and our Lamé mode resonator has been summarized in Table 1 in the revised manuscript. Our device has the advantage of low bias voltage and high Q factor.

  1. The English needs substantial grammatical polishing.

Our response:

Thank the reviewer for the helpful suggestion. The manuscript is carefully polished. 

Reviewer 2 Report

Dear authors,

Lamé modes are interesting test vehicles to investigate dissipation and material properties of single crystal silicon and the authors contribute by analyzing mechanical non-linearities.

A few comments:

  • The achieved fQ product -the figure of merit - under vacuum conditions is for this work is 1.5E12Hz, far below the most recent work that are at 2.8E13Hz for Silicon (Rodriguez et al. Sci. Rep., 2019) and even higher in Silicon Carbide (Hamelin et al., Sci. Rep., 2019) reaching 1E14Hz. Please discuss your work in view of these state-of-the-art results.
    • There are so few recent papers; only 2 from 2019... Please refer to more recent work especially if they are at the edge...
  • You basically use the HARPSS process and you do not provide any recent references related to their work. Comes to mind for example Daruwalla et al. Micro. Nano, 2020.
    • If your process is different from the HARPS process, clearly state the discrepancy.
    • Figure 4.b: the PECVD mask is in green as well as the 70nm sacrificial oxide --> please use a different color (violet for example)
    • In Figure 5, please show a cross-section image of the 70nm sacrificial gap
    • Are there any benefits to deposit Au on polySi ?
  • The motional impedance seems high for such a small gap; please confirm analytically the value Rm=300kΩ using (4)
  • I don't think your conclusions are correct. The mechanical non-linearity should not be observed when driving the resonator to sub-7nm displacement (given the gap is 70nm).
    • Other works in BAW Si resonators have shown vibration amplitude in the linear regime beyond 20nm.
    • Provide a concise write-up explaining your derivation
    • Units are not correctly written: N/m^(-3) should read N/m^3 or N·m^(-3)
    • Provide the fit and R-square for Figure 9
    • Provide the plot ∆f vs xmax^2 and the fit supporting lines 211 to 217.
    • Remove the nanoscale gaps and remeasure the mechanical vs electrostatic nonlinearity. If your conclusion holds, then you should still see mechanical nonlinearity at low displacements.
  • What is the value of Q_TED per Comsol simulations? Include the release holes in the simulation and the 4 anchoring tethers

Best regards,

Author Response

Response to the comments of the reviewers

First of all, we are grateful to the referees for their valuable works, which indeed help us improve the quality of our manuscript. We made the further revision to the manuscript according to their comments. All amendments are highlighted as red in the revised manuscript.

Reviewer 2:

Comments and Suggestions for Authors

Dear authors,

Lamé modes are interesting test vehicles to investigate dissipation and material properties of single crystal silicon and the authors contribute by analyzing mechanical non-linearities.

A few comments:

  1. The achieved f Q product -the figure of merit - under vacuum conditions is for this work is 1.5E12Hz, far below the most recent work that are at 2.8E13Hz for Silicon (Rodriguez et al. Sci. Rep., 2019) and even higher in Silicon Carbide (Hamelin et al., Sci. Rep., 2019) reaching 1E14Hz. Please discuss your work in view of these state-of-the-art results.

Our response:

Thank the reviewer for the helpful suggestion.

Several state-of-the-art high performance resonators were compared in table 1 of the revised manuscript.

The figure of merit of this work is inferior than the reported record-breaking silicon resonator (Rodriguez et al. Sci. Rep., 2019), however, the advantage of this work is achieving quite high fQ product at very low bias voltage (3V). The MEMS resonator with low bias voltage is highly desired for low power CMOS integrate system.  

  1. There are so few recent papers; only 2 from 2019... Please refer to more recent work especially if they are at the edge...

Our response:

Thank the reviewer for the helpful suggestion.

More recent works have been added as references and several related works are compared with our resonator in Table 1 in the revised manuscript.

  1. You basically use the HARPSS process and you do not provide any recent references related to their work. Comes to mind for example Daruwalla et al. Micro. Nano, 2020.

If your process is different from the HARPS process, clearly state the discrepancy.

Our response:

Thank the reviewer for the helpful suggestion.

The SOI based fabrication process in this work was developed based on our previous poly-silicon process for resonator [i]. The major discrepancies from HARPSS include:

  1. the gap refill and electrode shaping were realized by deposition and successively etching on the same polysilicon layer in this work, while the gap refill and electrode shaping were realized with two different polysilicon layers, that is, deposition, etching, and CMP of the first polysilicon layer, then deposition and etching the second polysilicon layer in HARPSS process (Daruwalla et al. Micro. Nano, 2020).
  2. The grounding structure is fabricated with heavily doped polysilicon layer and ensures extremely low feedthrough signal in the device, this is a specific process.
  3. The release process for the resonator based on HF solution is originally developed by our group [ii].
  1. Wei Luo, Hui Zhao, Quan Yuan, et al. A novel digitally tunable RF-MEMS disk resonator. Microsystem Technologies, 2017.
  2. Liu Y, Xie J, Zhao H, et al. An effective approach for restraining electrochemical corrosion of polycrystalline silicon caused by an HF-based solution and its application for mass production of MEMS devices. Journal of Micromechanics and Microengineering, 2012, 22(3):035003.

  1. Figure 4b: the PECVD mask is in green as well as the 70nm sacrificial oxide --> please use a different color (violet for example)

Our response:

Thank the reviewer for the helpful suggestion.

The figures are improved for better visual effect. The PECVD mask in Figure 4 was shown in violet in the revised manuscript.

  1. In Figure 5, please show a cross-section image of the 70nm sacrificial gap

Our response:

Thank the reviewer for the helpful suggestion.

The SEM figure of the 70 nm gaps has been added in Figure 5b, which shows the gap between the electrode and the vibrating plate.

  1. Are there any benefits to deposit Au on polySi ?

Our response:

Thank the reviewer for the question.

To achieve low resistance and high adhesivity, Cr/Au layers were deposited on polysilicon layers as electrode, which is beneficial for high frequency signal transmission. Considering the cost, other metals such as Al are acceptable substitution.

  1. The motional impedance seems high for such a small gap; please confirm analytically the value Rm=300 kΩ using (4)

Our response:

Thank the reviewer for the helpful suggestion.

Motional impedance is calculated and confirmed. The insertion loss of the transmission is     -75.38 dB, according to , the calculated motional impedance Rm = 293.8 kΩ. It is confirmed that the measured motional impedance is in accordance with analytical result, which are compared in Table 2.

It is likely that the resonator with nanoscale gap would have much smaller motional impedance. However, according to Equ. (9), the thickness of the vibrating plate is 2 μm, which leads to much smaller transduction area and lower bias voltage of 3 V, but relatively higher motional impedance than other resonators. More work will be done to improve the motional resistance by increasing the thickness of the vibrating plate and decreasing the transduction gap.

  1. I don't think your conclusions are correct. The mechanical non-linearity should not be observed when driving the resonator to sub-7nm displacement (given the gap is 70nm).

Other works in BAW Si resonators have shown vibration amplitude in the linear regime beyond 20nm.

Provide a concise write-up explaining your derivation

Our response:

Thank the reviewer for the helpful suggestion.

Carefully reconsideration was carried out on the nonlinearity of the resonator with nanoscale transduction gap. For capacitance resonators, the electrostatic coupling introduces nonlinear forcing terms, which takes a general form as [i]:

                                (1)

The conclusion in the manuscript is drawn from the comparison of calculated k3e and k3. However, for nano-scale spacing gap, there could be certain discrepancies between the theoretically calculated k3e and the actual one. It should be addressed that the spacing gap g is assumed to be 70 nm. Nevertheless, the effective gap size for the parallel plate assumption may deviate from the assumed 70 nm [ii]. Since k3e is inversely proportional to g5, a slight variation of g can induce significant k3e change. Thus, the actual k3e is very likely to be substantially higher than the calculated one. Therefore, for resonator nano-scale gaps, the small displacement of the device could make the electrical nonlinearity surpass the mechanical nonlinearity and results in spring soften. The corresponding expressions have been modified and marked red in the revised manuscript.

  1. Kaajakari V, Mattila T, Oja A, et al. Nonlinear limits for single-crystal silicon Microresonators. Journal of Microelectromechanical Systems, 2004, 13(5):715-724.
  2. Yang, Y., Ng, E. J., Hong, V. A., Ahn, C. H., Chen, Y., Ahadi, E., ... & Kenny, T. W. (2014, June). Measurement of the nonlinear elasticity of doped bulk-mode MEMS resonators. In Proc. Solid-State Sensors, Actuat., Microsyst. Workshop (pp. 8-12).

  1. Units are not correctly written: N/m^(-3) should read N/m^3 or N·m^(-3)

Our response:

Thank the reviewer for the helpful suggestion.

Sorry for the mistake. The correction has been done in the revised manuscript and marked in red.

  1. Provide the fit and R-square for Figure 9

Our response:

Thank the reviewer for the helpful suggestion.

The fit for Figure 9 is f(VDC) = sqrt(2671.87872-0.01032× VDC2), and the R-square is 0.9685, these have been added in Figure 9, explanation were added in the revised manuscript and marked in red.

  1. Provide the plot ∆f vs xmax^2 and the fit supporting lines 211 to 217.

Our response:

Thank the reviewer for the helpful suggestion.

The plot ∆f vs xmax^2 has been added in Figure 10.

  1. Remove the nanoscale gaps and remeasure the mechanical vs electrostatic nonlinearity. If your conclusion holds, then you should still see mechanical nonlinearity at low displacements.

Our response:

Thank the reviewer for the helpful suggestion.

This suggestion provides a reasonable routine to verify the discussions on mechanical vs electrostatic nonlinearity.  New devices with different gaps will be designed to study on the mechanical and electrostatic nonlinear behavior of Lamé-mode resonator, more credible progress will be reported in the near future.

  1. What is the value of Q_TED per Comsol simulations? Include the release holes in the simulation and the 4 anchoring tethers

Our response:

Thank the reviewer for the helpful suggestion.

The QTED we extracted from Comsol simulations with the structure including the release holes and anchoring tethers is about 3.6×105.

Round 2

Reviewer 2 Report

Dear authors,

Please complete the state-of-the-art table with the following recent and high ƒ·Q silicon Lamé mode resonators references:

• Daruwalla, A., Wen, H., Liu, C. et al. Low motional impedance distributed Lamé mode resonators for high frequency timing applications. Microsyst. Nanoeng. 6, 53 (2020)

• J. Yang, B. Hamelin, and F. Ayazi, “Capacitive Lamé Mode Resonators in 65 um-Thick Monocrystalline Silicon Carbide with Q-Factors Exceeding 20 Million,” in 2020 IEEE 33rd International Conference on Micro Electro Mechanical Systems (MEMS), Jan. 2020, pp. 226–229

• J. Rodriguez, S. A. Chandorkar, C. A. Watson, G. M. Glaze, C. H. Ahn, E. J. Ng, Y. Yang, and T. W. Kenny, “Direct Detection of Akhiezer Damping in a Silicon MEMS Resonator,” Scientific Reports, vol. 9, no. 1, pp. 1–10, Feb. 19, 2019.

• G. Xereas and V. P. Chodavarapu, “Wafer-Level Vacuum-Encapsulated Lamé Mode Resonator With f-Q Product of 2.23E13 Hz,” IEEE Electron Device Letters, vol. 36, no. 10, pp. 1079–1081, Oct. 2015.

In that table please add a column for fQ as well as a column for the capacitive gap so that your work can be easily distinguished from others. The SEM of the 70 nm gap is at an oblique angle and does not show the 70nm gap really but the sidewall roughness instead which looks important.

Please include an SEM heads on the gap. An example of such picture I am looking to get from you can be found in Daruwalla, A., Wen, H., Liu, C. et al. Low motional impedance distributed Lamé mode resonators for high frequency timing applications. Microsyst. Nanoeng. 6, 53 (2020) Figure 7. g

Further this roughness is significant and will impact the measure of k3e as the authors have noted. I appreciate that the claims about mechanical nonlinearity being dominant has been removed. Please include a table where you exhibit the values of Go, G1 and G2 that you obtain from your fit using k1, k2 and k3. Compare these values with recently reported values. Most importantly, more details on the grounding structure is needed if this is the main difference with HARPSS as I do not find any explication in your paper.

Best regards,

Author Response

Response to the comments of the reviewers

First of all, we are grateful to the referees for their valuable works, which indeed help us improve the quality of our manuscript. We made the further revision to the manuscript according to their comments. All amendments are highlighted as red in the revised manuscript.

Reviewer 2:

Comments and Suggestions for Authors

Dear authors,

  1. Please complete the state-of-the-art table with the following recent and high ƒ·Q silicon Lamé mode resonators references:
  • Daruwalla, A., Wen, H., Liu, C. et al. Low motional impedance distributed Lamé mode resonators for high frequency timing applications. Microsyst. Nanoeng. 6, 53 (2020)
  • J. Yang, B. Hamelin, and F. Ayazi, “Capacitive Lamé Mode Resonators in 65 um-Thick Monocrystalline Silicon Carbide with Q-Factors Exceeding 20 Million,” in 2020 IEEE 33rd International Conference on Micro Electro Mechanical Systems (MEMS), Jan. 2020, pp. 226–229
  • J. Rodriguez, S. A. Chandorkar, C. A. Watson, G. M. Glaze, C. H. Ahn, E. J. Ng, Y. Yang, and T. W. Kenny, “Direct Detection of Akhiezer Damping in a Silicon MEMS Resonator,” Scientific Reports, vol. 9, no. 1, pp. 1–10, Feb. 19, 2019.
  • G. Xereas and V. P. Chodavarapu, “Wafer-Level Vacuum-Encapsulated Lamé Mode Resonator With f-Q Product of 2.23E13 Hz,” IEEE Electron Device Letters, vol. 36, no. 10, pp. 1079–1081, Oct. 2015.

In that table please add a column for fQ as well as a column for the capacitive gap so that your work can be easily distinguished from others.

Our response:

Thank the reviewer for the helpful suggestion.

Those suggested state-of-the-art resonators have been added in Table 1, the fQ and the capacitive gap were also appended in the Table 1 of the revised manuscript.

  1. The SEM of the 70 nm gap is at an oblique angle and does not show the 70nm gap really but the sidewall roughness instead which looks important.

Please include an SEM heads on the gap. An example of such picture I am looking to get from you can be found in Daruwalla, A., Wen, H., Liu, C. et al. Low motional impedance distributed Lamé mode resonators for high frequency timing applications. Microsyst. Nanoeng. 6, 53 (2020) Figure 7. g

Our response:

Thank the reviewer for the helpful suggestion.

The SEM picture heads on the gap has been added in Figure 5b of the revised manuscript.

  1. Further this roughness is significant and will impact the measure of k3e as the authors have noted. I appreciate that the claims about mechanical nonlinearity being dominant has been removed. Please include a table where you exhibit the values of Go, G1 and G2 that you obtain from your fit using k1, k2 and k3. Compare these values with recently reported values.

Our response:

Thank the reviewer for the helpful suggestion.

The nonlinear shear modulus Go, G1 and G2 can be extracted from material nonlinearity using Eqn. 22 in the revised manuscript. In symmetrical structures, such as the Lamé-mode resonator presented in this work, k2 can be ignored, so that G1 is considered to be 0. For resonators with small gaps, the electrical nonlinearity cannot be ignored, so that the accurate G2 should be obtained after removing the electrical nonlinearity part in k3. The table below shows the extracted Go and G2 values of this work and the comparison with the reported works:

G0 (´1010 Pa)

|G2| (´1011 Pa)

Zhu, H.

5.11

2.85

Shao, L. C.

8.73

1.85

Yang, Y.

4.62

8.76

This work

6.99

4.03

The corresponding discussions have been added and marked red in the revised manuscript.

  1. Most importantly, more details on the grounding structure is needed if this is the main difference with HARPSS as I do not find any explication in your paper.

Our response:

Thank the reviewer for the helpful suggestion.

Detailed introduction about the grounding structure has been added in the revised manuscript and marked in red.
